# Impact of tourism on habitat use of black grouse *(Tetrao tetrix)* in an isolated population in northern Germany

**Daniel Tost**[1], **Egbert Strauß**[1], **Klaus Jung**[2], **Ursula Siebert**[1]*

**1** Institute for Terrestrial and Aquatic Wildlife Research, University of Veterinary Medicine Hannover, Foundation, Hannover, Germany, **2** Institute for Animal Breeding and Genetics, University of Veterinary Medicine Hannover, Foundation, Hannover, Germany

* ursula.siebert@tiho-hannover.de

**Data Availability Statement:** All relevant data are within the manuscript and its Supporting Information files.

**Funding:** The project was funded by the Ministry of Food, Agriculture and Consumer Protection of Lower Saxony (http://www.ml.niedersachsen.de).

## Abstract

As many other wild living species, black grouse *Tetrao tetrix* has to cope with anthropogenic disturbances in many habitats. Impacts of tourism and outdoor recreation on grouse species *Tetraoninae* have been subject to several studies in mountainous habitats in Central Europe and the United Kingdom. The geographically isolated and critically endangered black grouse population of Lower Saxony (North Germany) has been consistently monitored but beyond that poorly studied. This also applies to the key habitats of the nature reserve Lüneburg Heath (Lüneburger Heide) which, in turn, serves as a recreational area as well. In this study, the impact of tourism activity on habitat use of black grouse was investigated using data of GPS-tracked individuals. Additionally, visitor numbers on public and (usually undisturbed) closed routes were monitored using infrared light barriers. The spatio-temporal distribution of locations and the recreational activity were evaluated by linear mixed-effects models. Tagged individuals avoided the vicinity of public routes and avoiding distances were directly related to intensity of human activity. There was no seasonal change of black grouse habitat use alongside public routes. However, towards closed routes, significantly higher distances appeared during peak phases of visitor numbers (August and September), implying temporary increased disturbance levels within a key refuge area. Diurnal adaptation of habitat use was strongly dependent on the route density within the home range. Individuals used the vicinity of public trails at night and dawn but evaded these habitats during peak human activity around noon and afternoon. Recreational disturbances appeared to significantly affect the effective habitat availability for black grouse in the nature reserve. Visual cover by vegetation, however, seemed to diminish negative effects emerging from hiking trails. This provides an effective protective measure which requires minimal effort for the local conservation management.

This publication was supported by Deutsche Forschungsgemeinschaft (https://www.dfg.de) and University of Veterinary Medicine Hannover, Foundation (https://www.tiho-hannover.de) within the funding programme Open Access Publishing. The funders had no role in study design, data collection and analysis, decision to publish, or preparation of the manuscript.

**Competing interests:** The authors have declared that no competing interests exist.

## Introduction

The black grouse (*Tetrao tetrix*) suffered a severe decline within the Central European Lowlands during the 20[th] century [1]. In the German federal state of Lower Saxony, black grouse populations occurred across greater parts of the North German Lowlands until the 1950s with an estimated number of 7,000 to 9,000 individuals in 1959 [2]. The number then dropped to about 1,000 individuals in 1979, while most subpopulations went extinct due to large-scale habitat loss as a result of widespread conversion of heathland, moor and peat bogs to forest, farmland and pasture [3, 4]. Since the beginning of the annual recordings of black grouse populations by the Lower Saxony Federal Ornithological Station in 1995, numbers ranged around 200 individuals [5]. By now, the region Lüneburg Heath, which is located in eastern Lower Saxony, houses the last remaining population with only 130 confirmed individuals left in spring 2019 (Sandkühler, pers. comm.). It is considered the last autochthonous population within the Central European Lowlands and dispersed among five special protection areas (SPA) which are part of the European protected areas network NATURA 2000 [5]. Four of these sites are used as military training areas or firing ranges. The fifth and northern most site is the nature reserve Lüneburg Heath, which simultaneously serves as a local recreational area in the metropolitan region of Hamburg. As of 2019, only 30 individuals were confirmed on this site, this being the lowest number since 2000 (previous data: [5], recent data: Sandkühler, pers. comm.).

Both, the conservation of its natural and cultural elements and its function as a recreational area, are provided as resources worthy of protection in the nature reserve's regulations [6]. However, the execution advises on the conservation of breeding bird species of Lower Saxony as part of the "Strategy for Species and Biotope Conservation in Lower Saxony" which names disturbances of breeding grounds induced by recreational activities as one major concern for conservation objectives amongst others such as further habitat loss, predation pressure and isolation of subpopulations [7]. Regarding anthropogenic disturbances, this strategy paper advises the protection and quietening of display-, breeding- and rearing-grounds by establishing visitor guidance and visitor information [7].

On a global scale, the impact of tourism and recreation has been subject to numerous studies on a broad spectrum of wildlife species, providing broad findings of avoiding behaviour, adaptation of spatio-temporal habitat use and activity patterns due to human disturbances for several mammal species, including both predator and prey species [8–13].

Impacts such as winter sports, hiking, mountain-biking and related infrastructures cause well known negative effects on black grouse and its close relative capercaillie (*Tetrao urogallus*), resulting in fragmentation, habitat reduction or habitat avoidance [14–19]. Further described effects are reduction in abundance or activity of displaying black grouse cocks [15, 20], physiological stress [14, 21, 22] and increased flushing distances [23–25]. Consequential behavioural changes (e.g. flight) may not only affect body condition but also induce higher predation risk [21, 26]. Despite the fact that no effect on reproduction rates due to increased disturbance levels was found in black grouse [24] or capercaillie [19], there is evidence of negatively affected parental care and malnutrition of fledglings or juveniles in other bird species [27–29]. However, the actual impact of tourism on black grouse behaviour, physical condition and population dynamics in the Lüneburg Heath remains mostly unknown.

Given these findings and the local population's current critical condition, we took into further account how far black grouse are affected by outdoor activities such as hiking on the trail network of the nature reserve Lüneburg Heath. Our study therefore addresses the following questions: 1) Do black grouse avoid the proximity to hiking trails in general? 2) Is there an avoidance of the proximity to closed routes and trails? 3) Does the distance of black grouse

individuals to trails depend on the intensity of recreational use? 4) Do black grouse alter temporal habitat use due to hiking activity?

This study aimed to fill gaps in our current knowledge about the extent of the impact of recreational activities on this strictly protected species. Its results may contribute to a better understanding of spatial and temporal animal behaviour in adversely affected habitats and to future regional conservation concepts.

## Materials and methods

### Study area

The study was conducted within the nature reserve Lüneburg Heath in Lower Saxony, Germany (S1 Fig). It was first put under protection in 1922 (renewed and extended in 1993) and therefore is the second oldest nature reserve in Germany [30]. Today, it covers an area of about 23,440 ha of which 66% are forest, 22% heath and grassland, 6% farmland, 5% pasture and 1% routes, buildings and waterbodies [5, 30, 31]. It also contains about one quarter of the northern German black grouse population [5]. A peak number of 78 individuals (45 cocks; 33 hens) was counted within the nature reserve in 2007 during annual censuses conducted by the foundation Stiftung Naturschutzpark Lüneburger Heide and the Lower Saxony Federal Ornithological Station. Since then, numbers have been falling with 66 birds (38; 28) counted in 2011, 53 (25; 28) in 2015 [5] and 30 (14; 16) in 2019, respectively (Sandkühler, pers. comm.).

The study area covers 29 km$^2$ and contains two study sites (Fig 1). Site 1 is located on the north-western hillside of the Wilseder Berg, named after the nearby village Wilsede (53.166405° N, 9.961346° E), and with 169 m above sea level the highest elevation of the northwest German Lowlands. Site 2 is located in the opposite direction east of Wilsede and includes the Radenbachheide. Both sites predominantly consist of open heath and nutrient-poor grassland with small scattered shrubs of juniper or pioneer vegetation and are surrounded by dense pinewoods. Site 2 additionally contains extensive pastures.

The nature reserve's heathlands are the remains of a historic agricultural landscape development. From the Middle Ages until the end of the 19th century, sheep farming mixed with little arable farming for which organic topsoil from the heathland was extracted and used as fertiliser (i.e. plaggen farming) were the typical cultivation system in the glacially shaped landscape [30, 32]. It led to deterioration of forests and nutrient depletion in soils resulting in widespread heath vegetation surrounding the characteristic villages and farmyards [30, 32]. Nowadays, the preservation of open heathland is mainly realised by modern mechanical landscape maintenance and on a lower scale by heather burning and subsidised sheep farming [31].

Since the late 19th century, the historically established cultivation form has become increasingly unprofitable. Consequently, tourism and land use change towards reforestation gained in importance for economic survival. Old farmhouses were increasingly and still are used for gastronomic purposes as guesthouses and restaurants or as museums [30]. The nature reserve became more and more popular for its attractions of culture, landscape and natural assets [33]. Accordingly, the area has been highly frequented ever since the 20th century [30]. As of 1997, the annual number of visitors was estimated at four million visitors [30]. Thus, the nature reserve is accessible via a broad trail network, whereas trespassing offside designated public routes is prohibited [6]. Within the study area (29 km$^2$), the network contains 75.54 km of public routes and 6.48 km of closed routes, giving a route density of 2.6 km/km$^2$ and 0.22 km/km$^2$, respectively. Public routes are open for hiking, cycling, partly for horse riding and carriage rides but usually are closed to private car traffic. As part of a visitor guidance concept, chosen former public routes have been closed in order to relieve certain areas of human presence. These closed routes are marked with no trespassing signs (S1 Table; route IDs 3 and 4).

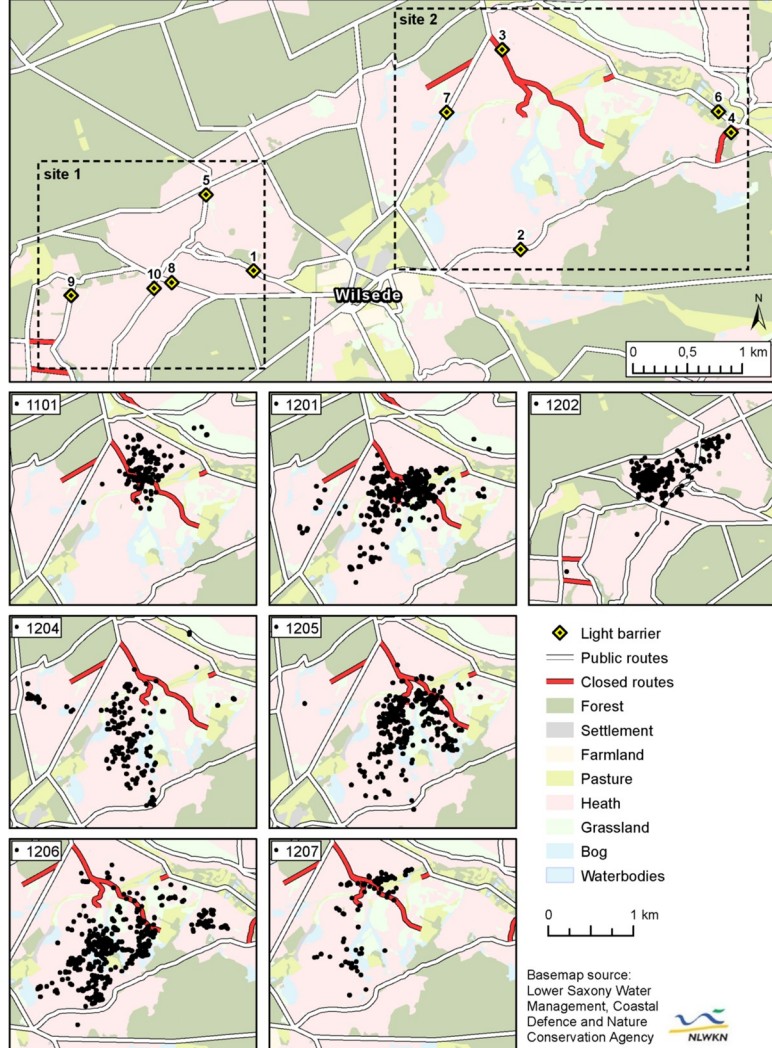

**Fig 1. Study area showing sites 1 and 2 and black grouse telemetry data.** Overview map (above) shows locations of light barriers alongside hiking trails and closed routes within the study sites 1 and 2. Detailed maps (below) show GPS locations of all seven tagged black grouse individuals.

Public and closed routes may, however, be used by local management vehicles for maintenance purposes and supply of livestock.

## Black grouse data

In 2011 and 2012, five cocks and two hens were caught in the eastern part of the nature reserve using stationary live traps. All birds were equipped with backpack mounted, battery operated GPS-tags with integrated VHF-module and accelerometers (e-obs GmbH, Gruenwald, Germany). Cocks were fitted with 38 g, hens with 28 g GPS-tags and tag-weight was kept below 3% of body weight (except for cock ID 1207 with 3.2%). One cock was tagged at site 1 in April 2012. The remaining six birds were tagged at site 2: one cock in May 2011, three cocks and both hens between March and May 2012 (Fig 1, Table 1). Individuals were located via VHF telemetry once a week to download the logged GPS locations from distances between 200 and 700 m. One hen (ID 1206) was recaptured and retagged in October 2012. Both hens

**Table 1. Summary of tagged black grouse individuals including number of GPS locations, date of capture and duration of data collection.**

| Animal ID | Sex | Age | Weight [g] | Number of locations | Home range [ha]; kernel 95% | Date tagged | Last position | Transmitting days | Study site | Individuals fate |
|---|---|---|---|---|---|---|---|---|---|---|
| 1101 | m | adult | 1304 | 159 | 39.05 | 08.05.2011 | 12.07.2011 | 64 | site 2 | remain unknown |
| 1201 | m | adult | 1361 | 452 | 132.99 | 25.03.2012 | 08.09.2012 | 167 | site 2 | preyed by goshawk |
| 1202 | m | adult | 1365 | 408 | 50.93 | 01.04.2012 | 10.09.2012 | 162 | site 1 | preyed, predator unknown |
| 1204 | m | adult | 1287 | 199 | 197.75 | 02.05.2012 | 02.07.2012 | 61 | site 2 | preyed by goshawk |
| 1205 | f | adult | 947 | 436 | 97.51 | 04.05.2012 | 03.12.2012 | 214 | site 2 | tag-battery exhausted, remain unknown |
| 1206 | f | adult | 991 | 546 | 192.31 | 06.05.2012 | 15.12.2012 | 223 | site 2 | preyed by fox or marten |
| 1207 | m | yearling | 1189 | 96 | 77.78 | 09.05.2012 | 30.07.2012 | 82 | site 2 | preyed by goshawk |

successfully hatched their clutches, of which one was replaced after loss to predation. Five birds were preyed by goshawks or other predators, one bird (ID 1101) went missing and could not be recovered and one bird's fate (ID 1205) remains unknown after the tag's battery was depleted in December (Table 1). Depending on survival rates and battery duration, numbers of transmitting days ranged between 61 and 223 days. GPS-locations were taken at predefined time intervals depending on the tags' programming, mostly every three hours between 01:00 and 10:00. In total 2,296 locations were taken. All stages of the animal experiment were conducted under a permit from the Lower Saxony Institute for Consumer Protection and Food Safety (LAVES, Dept. 33 Animal Welfare, permit number: 33.9-42502-04-11/0364). In order to minimise stress, handling was performed by a small, trained team in a quiet environment, while the animals' heads were covered.

### Visitor data

Between January 2015 and May 2017, the frequency of passing visitors (hikers, cyclists etc.) was continuously monitored using reflective infrared light barriers (Velleman NV, Gavere, Belgium) on ten routes, of which two were closed to the public (S1 Table). The chosen routes run along the edges or cross the home ranges of the seven test animals at sites 1 and 2 (Fig 1). Infrastructure of public and closed routes remained steady between 2011 and 2017. The light barriers and reflectors were hidden in fence posts with a typical appearance of the landscape and installed at a height of about 1.3 m. The systems batteries had a temperature-dependent durability of two to four weeks and were regularly exchanged. The posts were set up one to two metres apart along the waysides, so that the infrared beam covered the route's entire width. Date and time of each interruption of the infrared beam (trigger) were stored on Easy-Log data-loggers (OMEGA Engineering GmbH, Deckenpfronn, Germany). Trigger events may represent single persons, groups of people, cyclists and vehicles. Groups of persons may have been counted as one trigger if they had passed the infrared beam close to each other. Therefore, we considered each trigger as a potential disturbance event.

Before and after being used in the field, the light barriers were attached in a row and tested under controlled conditions. All light barriers triggered synchronously at walking speed and running speed. With the consent of the official data-protection supervisor of Lower Saxony, camera traps were installed next to selected light barriers for short sample periods in 2015 in order to verify the reliability of light barrier functioning in the field. The camera traps were operated during the calendar weeks 10, 15 and 19 along the routes ID 1 and 2. Camera trap images showed that on route 1 94% of the light barrier triggers were correct and 93% on route 2. Sheep herds and wildlife used the trails occasionally but did not trigger the light barriers as neither of them reached the height in which the light barriers were installed. Accordingly, wildlife and livestock could be excluded as significant triggers of light barriers.

Subsequently, the visitor datasets were cleared from technically flawed or obviously manipulated recordings. Series of false triggers due to loose contacts occurred once (route ID 7) and have been removed from the data set. The affected light barrier has been replaced.

## Handling data from different acquisition periods

Initially, it was planned to continue capturing, tagging and tracking of further black grouse individuals in 2015 and to simultaneously monitor visitor numbers. However, this telemetry study could not be realised after the annual monitoring indicated severely declined individual numbers in the nature reserve and a required permit was not granted by the responsible lower nature conservation authorities. Consequently, we used telemetry data of black grouse collected from 2011 to 2012 and visitor data from 2015 to 2017. For this reason, our analyses are based on the assumption that human activity and numbers of visitors remain about constant in the course of the year and day as well. We therefore extended the monitoring period over more than one year to evaluate the constancy of visitor numbers and route-specific frequency. This evaluation was carried out by correlating our monitoring data with freely accessible data of the Lower Saxony State Office for Statistics on overnight stays of tourists [34]. These data were available monthly aggregated for each year since 2009 and separately for all municipalities of Lower Saxony of which only municipalities of the nature reserve were considered hereafter (Undeloh, Schneverdingen, Egestorf; S2 Fig). We then compiled a matrix containing the mean numbers of light barrier triggers of each public route and the mean numbers of overnight stays of the three municipals separately for each year from 2009 to 2019. All mean values were aggregated for every month, respectively. Finally, we correlated the monthly mean values of each public route with the monthly mean values of overnight stays in all years using Spearman's rank correlation coefficient.

The correlation analysis confirmed our assumption as the monthly numbers of trigger events of all public routes, except route ID 6, were highly correlated with the monthly numbers of overnight stays of all years between 2009 and 2019 (S2 Table). For this reason, we found our visitor monitoring data to be comparable by months for different years and therefore evaluate the assumption that they were transferable for the period of black grouse tracking as reliable and reasonable.

## Data analysis

Across each home range of all seven tagged birds, Uniform random distributed points were generated using the feature class toolset "Create Random Points" in ArcGIS (version 10.1, ESRI). While the GPS locations represented the actual habitat use, the randomly generated points visualised a theoretical standard habitat use with neither preference nor avoidance of any structures. For both, GPS locations and randomly generated points, their direct distances to the nearest public and closed route were calculated. In the case of the cock 1202 (site 1), there were no closed routes nearby. Thus, no distances to closed routes were calculated in this instance.

Differences between distance distributions of random points and true GPS locations were descriptively analysed for both sexes, sites, closed and public routes separately and tested for significance using Wilcoxon's rank sum test in R [35]. The intensity of outdoor activities on routes was categorised as low, moderate, medium, high and very high according to the frequencies of passing visitors monitored between 2015 and 2017. The mean daily numbers of light barrier triggers in intervals of 20%-quantiles defined the categories (Table 2). The category low represented the closed routes, categories moderate to very high the public routes. Categorisation was necessary to ensure comparability of different monitoring periods (i.e.

**Table 2. Categorisation of visitors' occurrence by mean daily number of light barrier triggers.**

| Quantile | Average number of trigger events per day | Frequency of visitors (category) |
|---|---|---|
| < 20% | ≤ 17 | low (1) |
| 20–40% | 18–29 | moderate (2) |
| 40–60% | 30–40 | medium (3) |
| 60–80% | 41–67 | high (4) |
| > 80% | ≥ 68 | very high (5) |

black grouse data: 2011–2012 and visitor data: 2015–2017) as our analysis was based on the assumption that human activity remains approximately constant for several years. The distances of random points and GPS locations were assigned to the categories of the routes, respectively.

Linear mixed-effects models [36, 37] were used to compare distances of random points and true GPS locations separately for different categories of human activity. To account for repeated measurements, individuals were considered as random factors. In order to study the influence of human activity with temporal dependency, linear mixed models were fitted, again using individuals as random factors. The overall significance level for all tests and models was alpha = 5%.

## Results

### Human activity on public and closed routes

The visitor monitoring showed that the number of passing visitors varied between differently frequented public routes but kept a similar temporal distribution during the year as well as during the course of the day (Fig 2). In 2015 and 2016, we observed the lowest numbers during

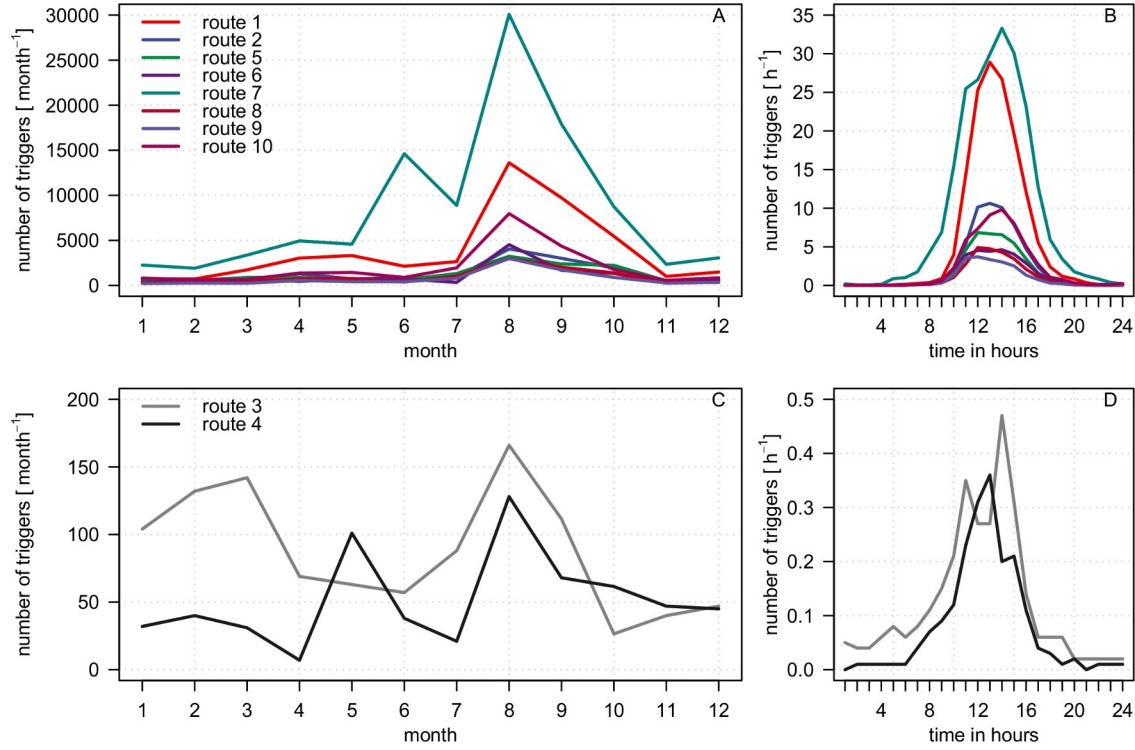

**Fig 2. Seasonal and diurnal human activity on public and closed routes.** Mean numbers of trigger events are averaged per month and per hour for public routes (A and B) and closed routes (C and D).

 

the winter months. During the blooming season of the heather in August and September, we observed an erratic increase in visitor numbers. Peak values were recorded on the main access route Wilseder Str. (route ID 7). Between October and November, the numbers dropped to the level of the winter months again. Compared to public routes, human activity was by far the lowest on closed routes and seasonally differed more due to the varying ratio of landscape management activity to trespassing visitors. With a mean number of 1.8 and 2.9 trigger events per day (S1 Table), both closed routes showed very low frequencies of human activity. In comparison, mean numbers on public routes ranged between 21.0 (route ID 9) and 225.7 triggers per day (route ID 7).

Considerable influxes of visitors on public routes began around 09:00 and 10:00, reached their maximum numbers between 12:00 and 14:00 and gradually subsided around 19:00. Only on the main access route Wilseder Str. did activity start earlier in the morning and last longer in the evening. On closed routes, the human activity fluctuated at a low level during daytime.

## Impact of public and closed routes on habitat use

Cocks and hens kept significantly higher distances towards public routes on site 2 than random points suggested (Wilcoxon rank sum test, males: p = 2.69e-22; females: p = 4.60e-26). This also applied to the cock on site 1 (p = 9.52e-41). Distances of locations and random points to public routes ranged between 0 and 800 m for both sexes within study site 2, but were differently distributed (Fig 3). Median distances of random points (cocks 316 m, hens 397 m)

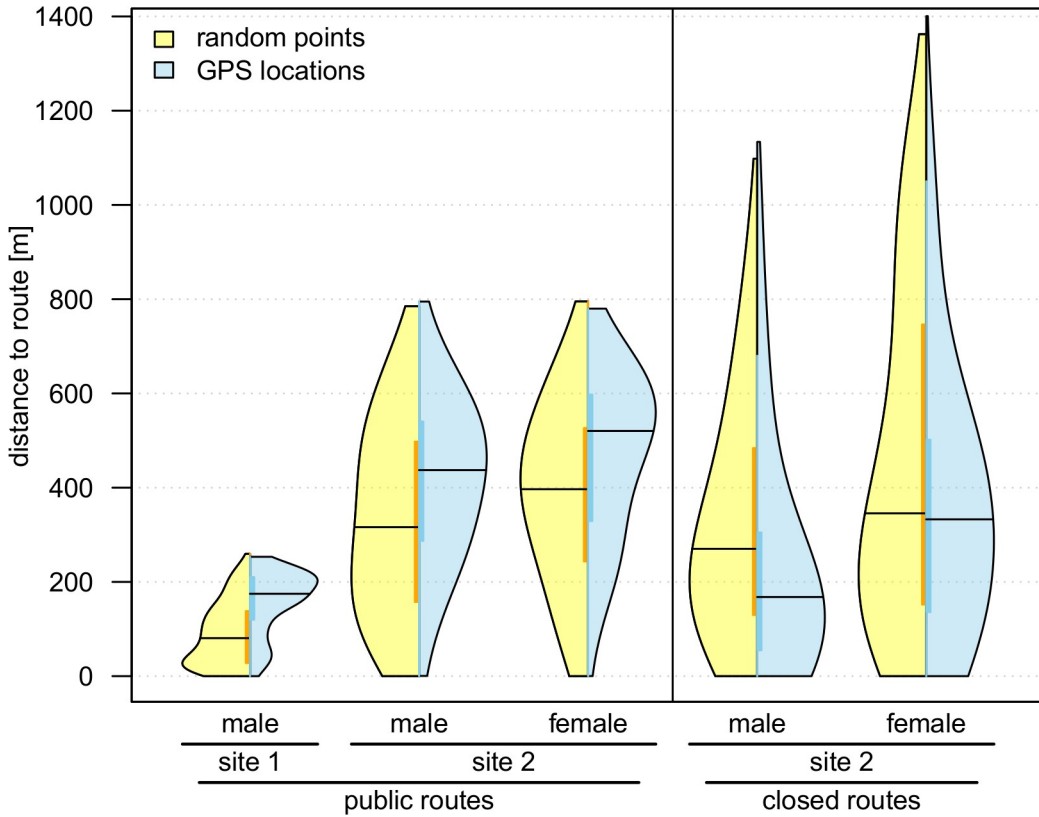

**Fig 3. Observed habitat use (GPS locations) versus expected habitat use (random points) alongside routes.** Violin plots show distributions of distances towards closest public and closed routes, respectively. Mean values are marked by black horizontal lines, quantiles are indicated by thick, coloured vertical bars. Violin bodies show the kernel density distributions of locations and random points over their distances to routes.

 

differed from median distances of both sexes' locations (cocks 437 m, hens 520 m). Equivalently the cock at study site 1 kept higher distances to public routes as well, while the distances only ranged between 0 and 253 m (median distances: 175 m for locations and 81 m for random points).

Regarding the distances to closed routes on site 2, cocks' GPS locations showed significantly lower distances than random points (median distances: 168 m for locations and 270 m for random points; p = 4.63e-23). Females' median distances of random points (346 m) and observed locations (333 m) to closed routes were quite similar. Distance distributions of females only varied in higher ranges (i.e. above 400 m), but still were significantly different in total (p = 4.97e-05).

## Impact of intensity of human activity

The comparison between distances of random points and GPS locations shows that individuals kept higher distances with increasing intensity of human activity (Fig 4). Moreover, individuals were less present in the vicinity of routes of moderate to very high intensity compared to routes of low intensity. Distance distributions of random points, however, showed higher variances between the categories of visitor frequencies (Table 2) and denser distributions in the vicinity of all routes regardless of the intensity of human activity. Avoidance behaviour was

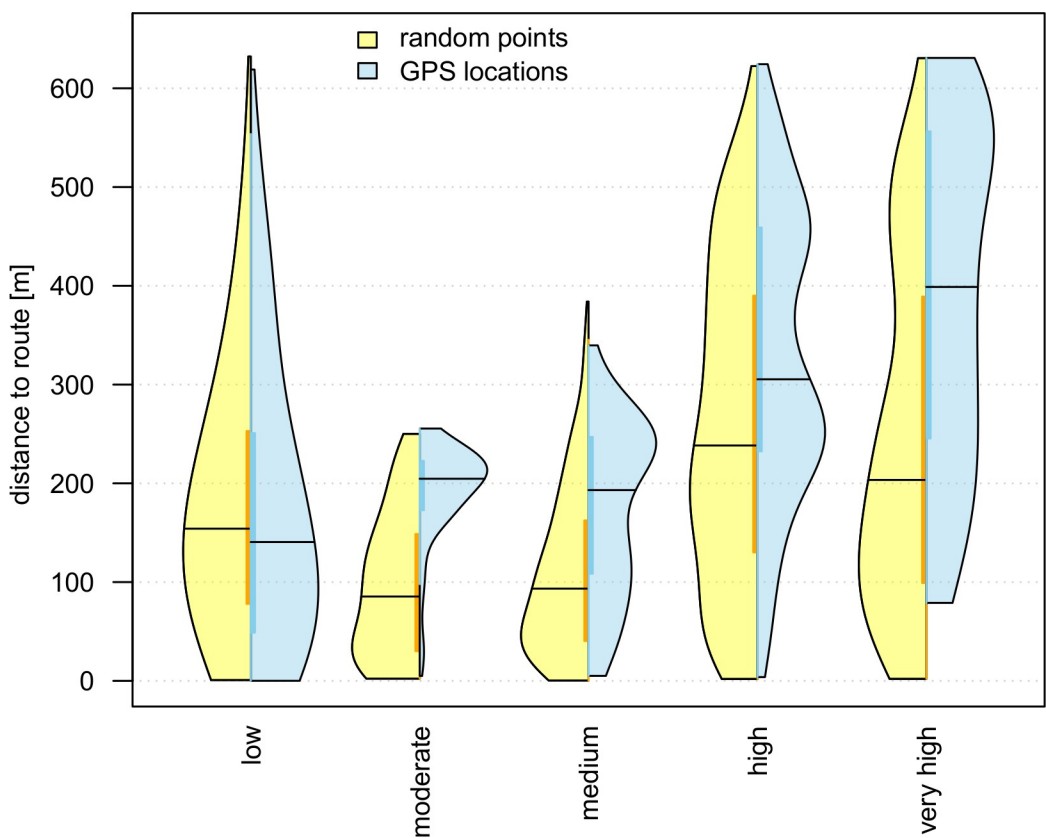

**Fig 4. Observed habitat use (GPS locations) versus expected habitat use (random points) in dependence of the intensity of human activity.** Distributions of distances towards routes are sorted according to category of visitor frequencies (from low to very high). Mean values are marked by black horizontal lines, quantiles are indicated by thick, coloured vertical bars. Violin bodies show the kernel density distributions of locations and random points over their distances to routes.

**Table 3. Linear mixed-effects model explaining distance of GPS locations and random points towards closest nearby routes depending on the intensity of human activity.**

| Variable | Estimate | SE | p-value | Sign. |
|---|---|---|---|---|
| Intercept | 125.14 | 18.73 | 0.0002 | *** |
| Intensity: random points | 23.12 | 1.52 | <2e-16 | *** |
| Intensity: GPS locations | 42.89 | 1.95 | <2e-16 | *** |

Intensity as categorised factors 1 to 5

well explained by the linear mixed-effects model with locations' distances towards routes increasing proportionally with frequencies of passing visitors. Based on a mean distance of 125.14 m towards routes, estimated distances of random points increased by 23.12 m per category level of human activity, and estimated distances of GPS locations nearly twice as much, by 42.89 m accordingly (Table 3).

## Temporal adaptation to human activity

There was no indication that individuals kept higher distances towards closed routes than expected in general. However, according to a linear mixed-effects model, there was an exceptional time-dependent shift to higher distances in the course of the year in August and September, which temporally coincides with the heather bloom and the associated increase in visitor numbers (Table 4, S3 Fig). Nonetheless, regarding public routes, such seasonal effects did not significantly appear.

The cock on site 1 used the direct vicinity of public routes at night, in the morning and in the late evening, but shifted its locations to higher distances with rising numbers of visitors around noon (Fig 5). On site 2, the diurnal adaptation of habitat use turned out to be less distinct. Therefore, the distance distributions towards public routes varied more noticeably, especially regarding the minimum distances. However, a linear mixed-effects model found that time of day and the hourly human activity on public routes as interacting explanatory variables were significant in explaining the diurnal shifts of black grouse distance distributions on both sites (Table 5). Around noon, the individuals kept higher distances to public routes than at night and at twilight. Correlation of distance distribution and hourly human activity as explained by a linear mixed-effects model is visualised in Fig 6.

**Table 4. Linear mixed-effects model explaining distance of black grouse GPS locations towards closed routes in dependence of season and seasonal human activity.**

| Variable | Estimate | SE | p-value | Sign. |
|---|---|---|---|---|
| Intercept | 362.39 | 61.24 | 0.00009 | *** |
| month3: mactclo | -1.11 | 0.33 | 0.00093 | *** |
| month4: mactclo | -2.38 | 0.54 | 0.00001 | *** |
| month5: mactclo | -1.78 | 0.51 | 0.00056 | *** |
| month6: mactclo | -1.31 | 0.59 | 0.02851 | * |
| month7: mactclo | -0.31 | 0.41 | 0.43694 | |
| month8: mactclo | 1.18 | 0.22 | 8.9e-08 | *** |
| month9: mactclo | 3.49 | 0.38 | <2e-16 | *** |
| month10: mactclo | -0.07 | 0.91 | 0.93742 | |
| month11: mactclo | -1.37 | 0.88 | 0.12270 | |
| month12: mactclo | -3.21 | 0.95 | 0.00073 | *** |

mactclo: monthly activity on closed routes

There are several locations of the individuals 1204, 1205 and 1207 in the vicinity (below 150 m) of a highly frequented hiking trail (route ID 2) that were recorded during the peak of daily visitor numbers between 11:00 and 14:00 (Fig 5). In this area, the trail was densely covered by vegetation. Of the hen 1206, locations of less than 150 m in the same area were recorded between 08:00 and 10:00 during comparatively lesser but rising human activity.

## Discussion

### Impact of public and closed routes on habitat use

Hiking trails and other public routes are the major recreation infrastructure of the nature reserve Lüneburg Heath. Our results confirm our initial hypothesis of a general avoidance of such routes as a result of human disturbance. In previous studies, negative impacts of outdoor recreation and recreation infrastructure were well described for black grouse [15, 16, 18, 20–22] and capercaillie [14, 17, 19, 23] in Alpine and other mountainous habitats. Many of these focused on winter sports but some considered summer outdoor recreation as well [17–19]. Further studies on black grouse and ptarmigan (*Lagopus mutus*) were conducted in England and Scotland and supported such findings [24–26]. So do our results which add new insight into avoidance behaviour for the structurally most different habitats of the North German Lowlands.

Black grouse as a ground breeding species relies on spacious, undisturbed refuges of sufficient quality and quantity [15, 22, 38]. In our study, site 2 offered large refuges with low density of route network, while site 1 was structured vice versa. At both sites, we confirmed general and diurnal avoidance behaviour at public routes. However, at site 1, both were more pronounced which might be explained by the substantially limited capacity of refuge. However, except for the blooming season of the heather, the closed routes at site 2, which cross the central refuge, were not avoided.

### Impact of intensity of human activity

The activity of visitors on different routes depends on various parameters such as popularity due to scenic attractiveness, location, route guidance, accessibility from car parks and gastronomy. Main access routes are accordingly highly frequented and some may as well be used by external suppliers. Recreational activities predominantly consist of hiking followed by cycling and lastly by horse riding. Carriage rides are offered during the peak season, usually concentrating on a few dedicated routes. As hypothesised, we could confirm a correlation between the intensity level of hikers' activities and the avoidance distance of tagged individuals. However, there appears to be a bias in the spatial distribution and density of routes with different category levels of visitor frequency. This uneven distribution leads to a disproportional emphasis of site 1 in the categories moderate and medium and of site 2 in the categories high and very high. Considering this bias, the observed effect still remains applicable. In other words, the more a route was frequented by visitors, the greater distances individuals kept from them. This corresponds with findings for Scottish capercaillie [19] and Alpine black grouse [22]. Deliberately and in different time intervals, regularly flushed black grouse individuals showed higher flight distances at increased disturbance levels [24]. Similar observations were described for capercaillie [23]. It is likely that based on these, our findings show the long-term consequences of persistent human disturbances on spatial use of black grouse and eventually an effective reduction in habitat availability that similarly has been observed in comparable studies [16–18]. Without considering the habitat structure of a particular study site, the conclusion that a high distance of individuals' locations towards trails implied a general avoidance behaviour would be inaccurate [15, 18]. However, in our study area, neither apart from trails nor in

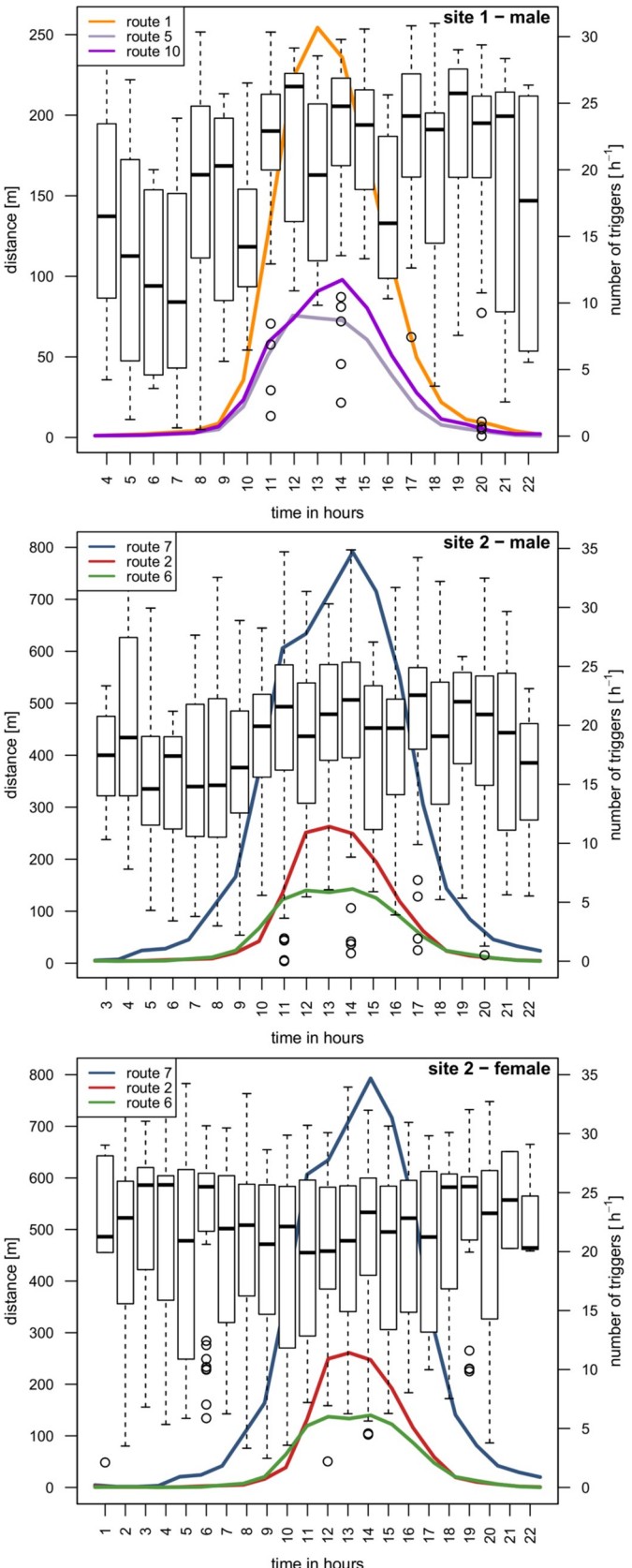

**Fig 5. Adaptation of black grouse habitat use to human presence by time of day.** Boxplots show distance distributions towards public routes in the course of the day (left-hand ordinate). Curves show occurrence of hikers on three trails closest to the home ranges (right-hand ordinate). Graphs are shown separately for sites and sexes. Telemetry and human activity data were collected in different monitoring periods. The comparison of both data sets is based on the assumption of constant visitor flows.

their vicinity did biotope structures differ considerably. We therefore conclude human activity to be a major variable to explain spatial avoidance alongside routes.

## Seasonal adaptation to human activity

Alongside closed routes, we observed a shift in the spatial distribution of individuals' locations towards higher distances only during the blooming season of the heather. The monitoring of human activity correspondingly showed increasing numbers on closed routes in August and September. Observation via camera traps confirmed an increase in illegal tourist activity in the same period. It also showed that light barriers were repeatedly and deliberately evaded. Human (illegal) activity on closed routes may therefore be underrated, whereas frequencies of vehicles for maintenance purposes remained constant.

Our model confirmed a temporal correlation between shift in black grouse habitat use and visitor activity. However, there may be further anthropogenic parameters that might induce seasonal habitat shifts which we did not test for. As in the rest of the nature reserve, the predominant open heathland areas at site 2 are subject to mechanical and traditional landscape conservation measures. As mechanical landscape maintenance measures are usually not conducted in late summer, we exclude these as a reason for the observed shifts. Moreover, over the entire year, the consistent regular presence of vehicles for supplying livestock seemed to be tolerated more than hikers. This coincides with previous studies, in which habituation of wildlife species to predictable disturbances has been widely discussed [8, 19, 23, 39–41].

Sheep grazing as a traditional form of landscape conservation may be able to affect black grouse habitat use. In northern England, reduced sheep grazing improved black grouse breeding success and increased population densities [42]. In the nature reserve Lüneburg Heath, sheep stocks in summer stay well around the reported tolerable value (i.e. < 1.1 sheep per ha [42]) [31]. However, in the nature reserve, sheep grazing is not bound to routes' vicinities and affects the heathland spaciously. We therefore do not expect that sheep grazing induced the observed shift in black grouse habitat use in this particular case.

Considering our observations, we argue that during the high season for tourist activity, increased trespassing becomes a relevant source of disturbance within usually undisturbed refuges. Consequently, the nature reserve's regulations on route restriction need to be enforced consistently, especially in ecologically sensitive phases and areas (i.e. concerning mating, breeding and rearing).Towards public routes, we could not detect seasonal shifts in black

**Table 5. Linear mixed-effects model explaining distance of black grouse GPS locations towards public routes in dependence of time of day and hourly activity.**

| Variable | Estimate | SE | p-value | Sign. |
|---|---|---|---|---|
| Intercept | 388.68 | 43.29 | 4.1e-05 | *** |
| night-time: hactpub | -30.99 | 14.71 | 0.0353 | * |
| twilight: hactpub | -8.58 | 2.90 | 0.0032 | ** |
| daytime: hactpub | 3.31 | 0.94 | 0.0130 | * |

hactpub: hourly activity on public routes

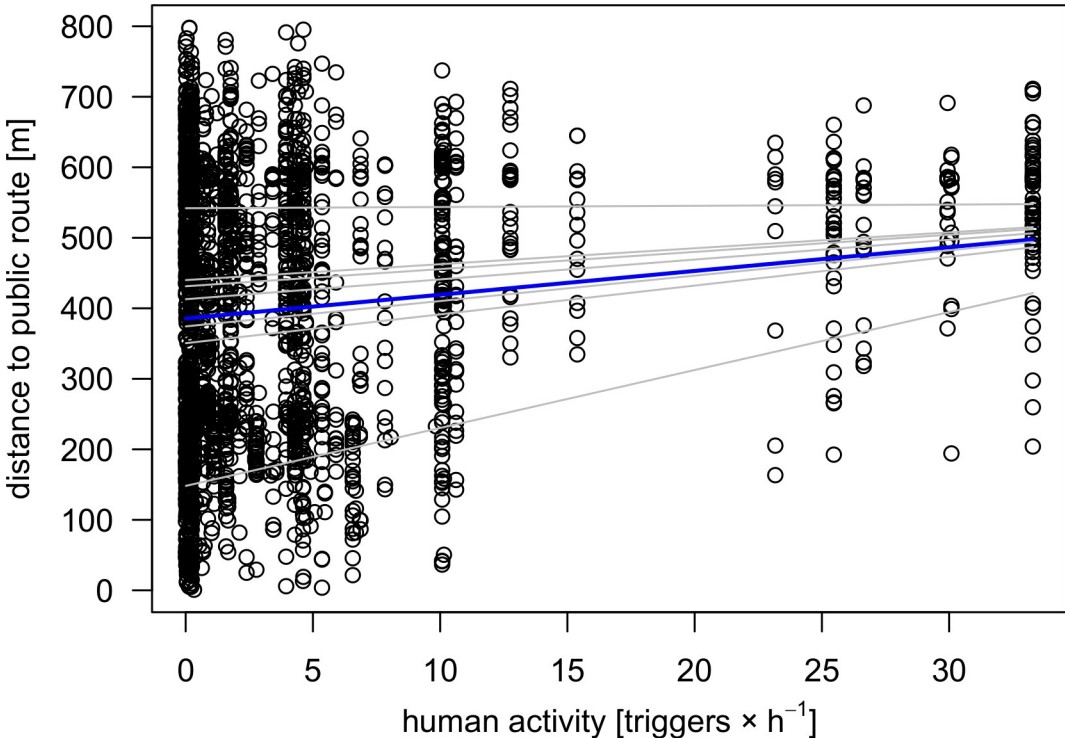

**Fig 6. Human activity in the course of the day affects distribution of black grouse locations.** The GPS locations (points) show the relationship between human activity and distribution of distances towards public routes. The linear mixed-effects model is plotted as a blue line; grey lines show the model's coefficients of each individual (random factor). Human activity on routes is calculated as average number of trigger events per hour. Telemetry and human activity data were collected in different monitoring periods. The comparison of both data sets is based on the assumption of constant visitor flows.

grouse habitat use. However, our data from tagged birds were only for the period from late March to mid-December. In the fourth quarter, only data of both hens were available after the males had died prematurely, indicating a high predatory pressure on the black grouse population. Thus, telemetry data was missing for the winter months when there is rather little human presence in the nature reserve. It remains unknown whether or not black grouse expanded their habitats closer towards public routes during months of relatively low recreational activity. Such behaviour might depend on the benefits of approaching the vicinity of routes (e.g. food availability). One reason for this may be that energetic cost of evasion of disturbances may be higher during winter [21] despite winters in northern Germany being relatively mild compared to the Alpine region. Nonetheless, combined with the reduced human presence, this may as well imply that black grouse seem more tolerant to disturbances in order to save energy [43]. However, this has yet to be investigated.

### Diurnal adaptation to human activity

The strong territoriality of male black grouse [38] might explain that despite persistent human disturbances the cock at site 1 maintained its home range and diurnally adapted its habitat use. Its avoidance of the routes' vicinity temporally correlated with the number of visitors, which is particularly high due to the local attraction, Wilseder Berg. Additionally, the route network at this study site is very dense, leaving little refuge areas within this individual's home range. Strategies of diurnal adaptation to human presence are well known for several mammalian wildlife species [8–13], including temporal shifts in activity patterns and spatial shifts in habitat use. We

only observed the latter in black grouse behaviour, while periods of activity apparently remained steady over the day. In fact, telemetry data and direct observations of the cock 1202 (site 1) confirmed it repeatedly displaying on a small burning site in the direct vicinity to an important public route in the early morning hours. Later in the morning, it withdrew towards the western hillside, which provided dense cover of vegetation and topography. In the Swiss Alps, abundances of displaying cocks were lower in recreational areas (e.g. ski lifts) [15, 20]. Yet, from our solitary observation of a displaying cock at site 1, we cannot draw conclusions on this site's suitability as a lekking ground in general. In contrast, males at site 2 displayed throughout the refuge, where they could hardly be observed from public routes. Lekking grounds located closest to public routes were located at distances of about 350 m but hidden behind the side of a valley.

Due to the characteristics of site 2 with wide refuge areas available and the individual home ranges of the tagged birds, diurnal adaptation of habitat use was less pronounced but still significantly verifiable. Both hens successfully hatched their clutches, of which one was placed near a closed route (route ID 3). The central area of site 2 may be a sufficiently undisturbed breeding area, but due to our sample size we did not investigate breeding success further. However, findings on reproduction success of ground breeding bird species under the influence of human disturbance present different results, where black grouse and capercaillie seem not to be affected during the hatching and raising of chicks [19, 24]. Nonetheless, ptarmigan showed impaired breeding success in recreational areas due to carrion crows (*Corvus corone*) following human development [26], scopoli's shearwater chicks (*Calonectris diomedea*) [27] and yellow-eyed penguin fledglings (*Megadyptes antipodes*) [44] gained less weight if exposed to higher disturbance levels and chick survival of snowy plovers (*Charadrius alexandrines*) [29] and hoatzins (*Opisthocomus hoazin*, not ground breeding) [28] were negatively affected by increased human recreational activities and tourism. For the moment, the actual impacts of recreational activity as well as other potential influences on reproduction success of black grouse in the nature reserve Lüneburg Heath remain uncertain. Regarding the decline in population size, we emphasise the need to continue investigations into this matter.

It is particularly noticeable that on site 2, distances below 150 m between 11:00 and 14:00 (Fig 5) refer to locations within an area of dense vegetation of juniper alongside a hiking trail of high intensity of human activity. In this area, hikers are hardly visible from the surrounding open heathland. This corresponds with several other studies highlighting the mitigating effects of visual cover on human disturbance impact and resulting habitat reduction [17, 18, 23]. Therefore, our results are of special significance for local landscape management plans. Providing sufficient visual cover along highly frequented routes serves the urgent necessity of quietening sensitive species' habitats [7] and may simultaneously be a cost-efficient protective measure of low impact on visitors' experience of nature. In certain circumstances (e.g. disturbed areas that provide spatially sufficient refuge), this may be an adequate alternative to drastic measures such as closure of established hiking trails or (temporal) areal exclusions of people [22, 23, 25] which might induce difficult conflicts of interests and need strict enforcement. Additionally, recommendations regarding management of protected areas depend on body size of targeted bird species, as larger birds appear to have greater alert distances and flight initiation distances [43]. In the case of larger bird species, this former study also advises to reduce the percentage of habitat open to visitors because higher spatial and temporal limitations on suitable habitat are to be expected for broadly accessible recreational areas [43].

### Management and research implications

The avoidance behaviour and preferential use of refuges show that black grouse in the nature reserve Lüneburg Heath are negatively affected by human disturbances but have so far been

able to compensate for the impact in its current state. The implementation of a detailed visitor guidance concept as one part of a large-scale nature conservation project was an important measure to harmonise the competing interests of species protection and recreational use of the nature reserve. In particular, the creation of a pattern of spacious refuges of high structural diversity throughout the open heathlands along with the closure or relocation of routes proved effective based on our results. Complementary to existing measures, we recommend to 1) further improve the connectivity between refuges, 2) provide visual cover by typical vegetation in sensitive habitats to reduce habitat loss and fragmentation, 3) foster understanding of the protective measures by local tourist education and improved signage and 4) strictly enforce the nature reserve's regulations in cases of trespassing.

Regarding the declining numbers found in population monitoring in recent years, we cannot evaluate the weight of tourist activity on population development compared to other, mostly unknown environmental parameters. This would require a different, more comprehensive approach [45] and inclusion of further methods (e.g. hormonal and genetic faecal sampling and documentation of dropping distribution [14, 18, 19, 23]). In the context of tourism and recreation, numerous previous studies consider further crucial impacts that our study did not focus on, such as physiological stress [14, 22], reduction in displaying activity (genetic diversity) [15, 20], increased predatory risk [26, 46], which may reveal multiple potential threats to the population development.

Future research on the isolated North German black grouse should be targeted more comprehensively in terms of study area and study aims. This should contribute to a better understanding of the species's and especially the population's vulnerability and eventually to improved conservation concepts. The nature reserve's western black grouse habitats should be involved as well as the neighbouring special protection areas (SPA) under the EU Birds Directive (military training areas south of the nature reserve), which contain the remaining three quarters of the Lower Saxonian black grouse population. The focus of research should include 1) the impact of predators on nests, fledglings and adults, 2) habitat suitability and food availability, including the insectivorous diet of chicks, and 3) the condition of migration and genetic exchange between key habitats on the scattered SPAs.

## Supporting information

**S1 Table. Duration of visitor monitoring, numbers of recorded trigger events of all ten light barriers and assigned categorisation of visitor frequency.**
(DOCX)

**S2 Table. Mean monthly numbers of light barrier triggers (years 2015–2017) and overnight stays in three municipalities (years 2009–2019) are highly correlated for all public routes except route ID 6.**
(DOCX)

**S1 Fig. Location of the study area. The nature reserve Lüneburg Heath and its composition of land use.**
(TIF)

**S2 Fig. Comparison of numbers of guest overnight stays (A) and light barrier trigger events (B) per month.** Overnight stays are monthly cumulated for three municipalities of the nature reserve Lüneburg Heath (Schneverdingen, Undeloh, Egestorf). Trigger events are monthly cumulated for each monitored public route.
(TIF)

**S3 Fig. Increasing visitor numbers on closed routes induce a shift of black grouse locations to higher distances towards closed routes.** Distance distributions are visualised as boxplots by month; red marks show the linear mixed-model's estimates of distance explained by monthly activity of visitors. Individuals were considered as random factors.
(TIF)

## Acknowledgments

We wish to thank the association Verein Naturschutzpark e.V. and the foundation Stiftung Naturschutzpark Lüneburger Heide for their energetic and logistic support of this project. We particularly thank M. Zimmermann, M. Sander, S. Wormanns and S. Weber. Furthermore, we also thank all persons who supported and realised our fieldwork, especially J. Hindersin, A. Niebuhr, F. Cecchini and the junior colleagues of the federal voluntary service. Moreover, we wish to thank K. Sandkühler of the Lower Saxony Federal Ornithological Station for contributing the numbers of annual black grouse counting. We would also like to thank O. Keuling, U. Voigt and R. Reding for their scientific support and F. Sherwood-Brock for language proofreading of the manuscript.

## Author Contributions

**Conceptualization:** Daniel Tost, Egbert Strauß.

**Data curation:** Daniel Tost.

**Formal analysis:** Daniel Tost.

**Funding acquisition:** Daniel Tost, Egbert Strauß, Ursula Siebert.

**Investigation:** Daniel Tost.

**Methodology:** Daniel Tost, Klaus Jung.

**Project administration:** Daniel Tost, Egbert Strauß.

**Resources:** Daniel Tost.

**Software:** Daniel Tost.

**Supervision:** Klaus Jung, Ursula Siebert.

**Validation:** Daniel Tost.

**Visualization:** Daniel Tost.

**Writing – original draft:** Daniel Tost.

**Writing – review & editing:** Daniel Tost, Egbert Strauß, Klaus Jung, Ursula Siebert.

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
