## [Decision Letter · Decision Letter 0]

30 Jun 2020

PONE-D-20-14230

Impact of tourism on habitat use of black grouse (Tetrao tetrix) in an isolated population in northern Germany

PLOS ONE

Dear Dr. Siebert,

Thank you for submitting your manuscript to PLOS ONE. After careful consideration, we feel that it has merit but does not fully meet PLOS ONE’s publication criteria as it currently stands. Therefore, we invite you to submit a revised version of the manuscript that addresses the points raised during the review process.

We look forward to receiving your revised manuscript.

Kind regards,

Bi-Song Yue, Ph.D

Academic Editor

PLOS ONE

2. We note that Figure 1 and S1 Fig in your submission contain map images which may be copyrighted.

a. You may seek permission from the original copyright holder of Figure 1 and S1 Fig to publish the content specifically under the CC BY 4.0 license. 

Reviewers' comments:

Reviewer's Responses to Questions

**Comments to the Author**

1. Is the manuscript technically sound, and do the data support the conclusions?

Reviewer #1: Partly

Reviewer #2: No

2. Has the statistical analysis been performed appropriately and rigorously? 

Reviewer #1: Yes

Reviewer #2: I Don't Know

3. Have the authors made all data underlying the findings in their manuscript fully available?

Reviewer #1: Yes

Reviewer #2: No

4. Is the manuscript presented in an intelligible fashion and written in standard English?

Reviewer #1: No

Reviewer #2: Yes

5. Review Comments to the Author

Reviewer #1: General Remarks

I have read with interest the study evaluating space use of on endangered bird population in relations to recreational pressure. The study is clearly original and to my knowledge has not been published elsewhere. The data at hand seem to have been evaluated correctly and support the conclusions soundly.

In general however, I think that the data set is limited as (1) a relative short period of bird activity (one season or 60 to 223 days) has been recorded and (2) data of human recreational activity was monitored in different time frame as the bird spatial data. The second point provides problems for some of the graphs (Fig 5 and 6) where both data sets are shown in direct relation to each other. Overall, I think that the hypothesis (=grouse avoids human presence) and the resulting conclusions are hardly surprising and do not contribute much to field per se. Despite this, the study contributes information to regional aspect of the species and is worth publishing despite the above shortcoming. As PLOS One is less concerned with overall impact on the scientific field, this alone might be not a reason to reject.

The text is written mostly in standard English but phrasing is awkward in places (see under specific remarks) and the manuscript is in need of overall language editing.

Specific remarks

Line 78: rephrase otherwise unclear. Maybe state it more directly: Does distance to trail depends on intensity of recreational use.

Line 79 Rephrase the sentence. Suggestion:

Do black grouse alter temporal habitat use due to hiking activity Or: Do b c avoid certain areas during times of increased hiking activity?

Line 89 strange statement. Please revise: A peak number of 78 individual was counted in 2007 during annual censuses conducted by ....

Line 106ff It was not only grazing that kept the forest from returning. Cutting and removing of organic soil swards (plaggen, see https://en.wikipedia.org/wiki/Plaggen_soil ) used to be the traditional use of heath landscapes. Results were reduced organic soil horizon and reduced nutrient availability. A good citation of this practice and its result for the local ecology would be: Leuschner C & Ellenberg, H. 2017 Vegetation Ecology of Central Europe

Line 130: Please add more details about the GPS loggers, were they battery operated, solar?

Line 145: Pleases revise sentence. It reads now as if the populations size went down because of the monitoring (I hope this was not the case).

Line 148: Is their information on the age and status of the birds available? Adult vs. first year at least should be possible. weight?

Line 166 Does that mean that no life stock nor larger wildlife used the trails at all? Can this be excluded?

Line 171: Please clarify: random points are not equally distributed

Line 235: Explain the graph a bit more, e.g. what does the thickness of the "violin body" stands for

Line 302: The different time [periods covered are indeed a major problem, especially for the analysis of data shown in Fig 5 and 6. I don’t think these graphs should be used for that reason. The use of categories and levels as use in the other graphs is appropriate.

Line 309 Can data analysis for this tested assumption be shown? In general, all this might be better placed under methods.

Reviewer #2: This paper has examined the impact of tourism on the habitat use of black grouse population in northern Germany.

However, the reviewers were unable to clarify the purpose of this manuscript and could not be convinced that there was sufficient evidence to support the views expressed in the discussion.

Reviewer would like the authers to reorganize the sentence and the data as a whole.

In addition, please address the following points.

L84-126: The historical or social background of the Study area should be explained in the “introduction” or “discussion”.

L129-150: Reviewer could not read the history of individual tracking, tagging, tag swapping, predation, etc. for each bird.

What does "Cause of loss" in Table 1 mean?

Does the "Transmitting days" difference affect the analysis of the data?

L162-164: Therefore, we considered each trigger as a potential disturbance event and as a relative instead of an exact number of visitors. What is mean the word “a relative”?　Please describe it specifically as "a cumulative total number" or describe the method of counting.

L152-168: In this study, it seems to have collected differently meaningful data. Each method and procedure should be explained separately.

1. Frequency of passive visitors counted with reflective infrared light barriers.

2. Verifying the reliability of light barrier functioning.

3. Handling technically flawed or obviously manipulated recordings.

L264: The word "the heather bloom" is only mentioned in L264, what does it mean?

6. PLOS authors have the option to publish the peer review history of their article (what does this mean?). If published, this will include your full peer review and any attached files.

Reviewer #1: No

Reviewer #2: **Yes: **Takeshi Kawasaki

---

## [Author Response · Author response to Decision Letter 0]

14 Aug 2020

Dear Bi-Song Yue, 

Thank you for providing the reviews on our manuscript “Impact of tourism on habitat use of black grouse (Tetrao tetrix) in an isolated population in northern Germany”. We have carefully adapted all of the reviewer’s suggestions and provide a detailed response to their points of criticism. We believe the manuscript has improved due to the review process and hope it is now suitable for publication in the journal of PLOS ONE.

We would like to thank the reviewers again for their valuable comments. Please find our detailed responses below. All authors have approved the manuscript and rebuttal letter. 

Kind Regards,

Daniel Tost,

Egbert Strauß,

Klaus Jung,

Ursula Siebert

Response to academic advisor (journal requirements):

- Title page has been changed to match the style requirements

- We shortened the titles of tables and added table legends where necessary

- We added one supplementary figure and one supplementary table. The order of the sup. figures has changed and figures have been relabeled accordingly.

2. We note that Figure 1 and S1 Fig in your submission contain map images which may be copyrighted.

- We added a copy of the original usage agreement and a translated version

Response to Reviewers:

Reviewer #1: 

General Remarks

I have read with interest the study evaluating space use of on endangered bird population in relations to recreational pressure. The study is clearly original and to my knowledge has not been published elsewhere. The data at hand seem to have been evaluated correctly and support the conclusions soundly. 

In general however, I think that the data set is limited as (1) a relative short period of bird activity (one season or 60 to 223 days) has been recorded and (2) data of human recreational activity was monitored in different time frame as the bird spatial data. The second point provides problems for some of the graphs (Fig 5 and 6) where both data sets are shown in direct relation to each other. Overall, I think that the hypothesis (=grouse avoids human presence) and the resulting conclusions are hardly surprising and do not contribute much to field per se. Despite this, the study contributes information to regional aspect of the species and is worth publishing despite the above shortcoming. As PLOS One is less concerned with overall impact on the scientific field, this alone might be not a reason to reject.

The text is written mostly in standard English but phrasing is awkward in places (see under specific remarks) and the manuscript is in need of overall language editing.

- The manuscript has been proof read by a native speaker (see acknowledgements). Specific remarks have been edited.

Specific remarks

1) Line 78: rephrase otherwise unclear. Maybe state it more directly: Does distance to trail depends on intensity of recreational use.

- Hypothesis has been rephrased without altering its meaning

2) Line 79 Rephrase the sentence. Suggestion:

Do black grouse alter temporal habitat use due to hiking activity Or: Do b c avoid certain areas during times of increased hiking activity?

- Hypothesis has been rephrased without altering its meaning; Suggestion no. 1 has been adopted

3) Line 89 strange statement. Please revise: A peak number of 78 individual was counted in 2007 during annual censuses conducted by....

- The sentence has been rephrased

4) Line 106ff It was not only grazing that kept the forest from returning. Cutting and removing of organic soil swards (plaggen, see https://en.wikipedia.org/wiki/Plaggen_soil ) used to be the traditional use of heath landscapes. Results were reduced organic soil horizon and reduced nutrient availability. A good citation of this practice and its result for the local ecology would be: Leuschner C & Ellenberg, H. 2017 Vegetation Ecology of Central Europe

- We added plaggen-farming as a landscape shaping practice. Thank you very much for the suggested reference which we inserted. It will also be very helpful for one of our upcoming publications. 

5) Line 130: Please add more details about the GPS loggers, were they battery operated, solar?

- We added more information on the GPS loggers

6) Line 145: Pleases revise sentence. It reads now as if the populations size went down because of the monitoring (I hope this was not the case).

- No, fortunately the monitoring had no adverse effect on the population. We changed the sentence accordingly and moved it to the section “Handling data from different acquisition periods” in order to avoid repetitive mentioning in the Materials part (see remark 12 of Reviewer #1).

7) Line 148: Is their information on the age and status of the birds available? Adult vs. first year at least should be possible. weight?

- We added information on age (adult, yearling) and weight in Table 1

8) Line 166 Does that mean that no life stock nor larger wildlife used the trails at all? Can this be excluded?

- Both, wildlife and livestock (only sheep herds) used the trails occasionally. Roe deer was the largest detected wildlife species and (as sheep) did not trigger the light barriers due to their height. We added the sentence: “Sheep herds and wildlife used the trails occasionally but did not trigger the light barriers as neither of them reached the height in which the light barriers were installed.”

9) Line 171: Please clarify: random points are not equally distributed

- You are right. The points were not regular (equidistant) distributed but drawn from a Uniform random distribution for both spatial axes. We changed the sentence: “Across each home range of all seven tagged birds, Uniform random distributed points were generated using the feature class toolset “Create Random Points” in ArcGIS …”, according to the description of the tool (https://desktop.arcgis.com/en/arcmap/10.3/tools/data-management-toolbox/how-create-random-points-works.htm)

10) Line 235: Explain the graph a bit more, e.g. what does the thickness of the "violin body" stands for

- Descriptions of Figures 3 and 4 have been stated more precisely

11) Line 302: The different time [periods covered are indeed a major problem, especially for the analysis of data shown in Fig 5 and 6. I don’t think these graphs should be used for that reason. The use of categories and levels as use in the other graphs is appropriate.

- Thank you for your comment on this matter. We have carefully considered this remark and think that, based on the verified assumption of the constancy of visitor flows, the use of the figures is justified. However, for better understanding, we will explicitly mention the different time periods in the captions of figures 5 and 6.

12) Line 309 Can data analysis for this tested assumption be shown? In general, all this might be better placed under methods.

- The section “Handling data from different acquisition periods” has been moved from Discussion to Material and methods. 

We added the analysis of our assumption concerning the constancy of visitor numbers over several years (see S2 Table and S2 Fig). Our analysis compares our monitoring data with independent data of tourist overnight stays in the region. Other than our data, the numbers of overnight stays are available for a longer period (2009-2019). Our analysis showed that our monitoring data is highly correlated with the overnight stays by month for all years.

Reviewer #2: 

This paper has examined the impact of tourism on the habitat use of black grouse population in northern Germany.

However, the reviewers were unable to clarify the purpose of this manuscript and could not be convinced that there was sufficient evidence to support the views expressed in the discussion.

- Referring to the statement of Reviewer #1 that “the study contributes information to regional aspect of the species and is worth publishing“, we added a brief concluding statement of our study’s overall aim and its intended contribution at the end of the introduction. We hope that this will help to clarify the purpose of our work for the readers and to dispel the reviewer’s doubts. 

Reviewer would like the authers to reorganize the sentence and the data as a whole.

- We could not fully understand the reviewer's desire for comprehensive restructuring. We have clarified or revised unclear and contentious sections and, in the event of further ambiguities, are open to specific suggestions, which we are happy to take into account in order to further improve the quality of the manuscript.

In addition, please address the following points.

1) L84-126: The historical or social background of the Study area should be explained in the “introduction” or “discussion”.

- We think that the description of the study area’s background fits best into the section “Study area” and would like to keep it as is. 

2) L129-150: Reviewer could not read the history of individual tracking, tagging, tag swapping, predation, etc. for each bird.

What does "Cause of loss" in Table 1 mean?

Does the "Transmitting days" difference affect the analysis of the data?

- We added information on tracking: “Individuals were located via VHF telemetry once a week to download the logged GPS locations from distances between 200 and 700 m.” 

Tagging has already been described (e.g. type of tag, tag weight, tag-mounting and handling of animals). Tags have been swapped only once on one individual: “One hen (ID 1206) was recaptured and retagged in October 2012.” No other individual has been retagged.

Regarding predation we added the sentence: “Five birds were preyed by goshawks or other predators, one bird (ID 1101) went missing and could not be recovered and one bird’s fate remains unknown after the tag’s battery was depleted in December (Table 1).” Detailed information on predation or remain of individuals can be found in Table 1. “Cause of loss” has been replaced by “Individual fate” in Table 1.

We took account of the difference of the transmitting days by using the individuals as random factors in our linear mixed effects models. 

3) L162-164: Therefore, we considered each trigger as a potential disturbance event and as a relative instead of an exact number of visitors. What is mean the word “a relative”?　Please describe it specifically as "a cumulative total number" or describe the method of counting.

- We deleted the half sentence “… and as a relative instead of an exact number of visitors.” and added a previous sentence “Groups of persons could have been counted as one trigger if they passed the infrared beam close to each other.”

4) L152-168: In this study, it seems to have collected differently meaningful data. Each method and procedure should be explained separately.

1. Frequency of passive visitors counted with reflective infrared light barriers.

2. Verifying the reliability of light barrier functioning.

3. Handling technically flawed or obviously manipulated recordings.

- We added information on the method of visitor monitoring and structured it as suggested. The frequencies of counted visitors are part of the Results and can be found in the section “Human activity on public and closed routes”.

5) L264: The word "the heather bloom" is only mentioned in L264, what does it mean?

- Heather bloom describes the event and season when the heather is in full bloom and additionally describes the associated seasonal attraction for visitors. Synonyms are heather blossom; blooming season of the heather.

---

## [Editor Report · Decision Letter 1]

21 Aug 2020

Impact of tourism on habitat use of black grouse (Tetrao tetrix) in an isolated population in northern Germany

PONE-D-20-14230R1

Dear Dr. Siebert,

We’re pleased to inform you that your manuscript has been judged scientifically suitable for publication and will be formally accepted for publication once it meets all outstanding technical requirements.

Kind regards,

Bi-Song Yue, Ph.D

Academic Editor

PLOS ONE

---

## [Editor Report · Acceptance letter]

26 Aug 2020

PONE-D-20-14230R1 

Impact of tourism on habitat use of black grouse *(Tetrao tetrix)* in an isolated population in northern Germany 

Dear Dr. Siebert:

I'm pleased to inform you that your manuscript has been deemed suitable for publication in PLOS ONE. Congratulations! Your manuscript is now with our production department. 

Kind regards, 

on behalf of

Dr. Bi-Song Yue 

Academic Editor

PLOS ONE